# Artificial Intelligence in Urooncology: What We Have and What We Expect

**DOI:** 10.3390/cancers15174282

**Published:** 2023-08-26

**Authors:** Anita Froń, Alina Semianiuk, Uladzimir Lazuk, Kuba Ptaszkowski, Agnieszka Siennicka, Artur Lemiński, Wojciech Krajewski, Tomasz Szydełko, Bartosz Małkiewicz

**Affiliations:** 1Department of Minimally Invasive and Robotic Urology, University Center of Excellence in Urology, Wroclaw Medical University, 50-556 Wroclaw, Poland; alina.semianiuk@student.umw.edu.pl (A.S.); uladzimir.lazuk@student.umw.edu.pl (U.L.); wojciech.krajewski@umw.edu.pl (W.K.); tomasz.szydelko@umw.edu.pl (T.S.); 2Department of Physiotherapy, Wroclaw Medical University, 50-368 Wroclaw, Poland; kuba.ptaszkowski@umw.edu.pl; 3Department of Physiology and Pathophysiology, Wroclaw Medical University, 50-556 Wroclaw, Poland; agnieszka.siennicka@umw.edu.pl; 4Department of Urology and Urological Oncology, Pomeranian Medical University, 70-111 Szczecin, Poland; artur.leminski@pum.edu.pl

**Keywords:** artificial intelligence, machine learning, urooncology, prostate cancer

## Abstract

**Simple Summary:**

Our study provides an overview of the current state of artificial intelligence applications in urooncology and explores potential future advancements in this field. With remarkable progress already achieved, artificial intelligence has revolutionized urooncology by facilitating image analysis, grading, biomarker research, and treatment planning. We also discuss types of artificial intelligence and their possible applications in the management of cancers such as prostate, kidney, bladder, and testicular. As artificial intelligence technology continues to evolve, it holds immense promise for further advancing urooncology and enhancing the care of patients with cancer.

**Abstract:**

Introduction: Artificial intelligence is transforming healthcare by driving innovation, automation, and optimization across various fields of medicine. The aim of this study was to determine whether artificial intelligence (AI) techniques can be used in the diagnosis, treatment planning, and monitoring of urological cancers. Methodology: We conducted a thorough search for original and review articles published until 31 May 2022 in the PUBMED/Scopus database. Our search included several terms related to AI and urooncology. Articles were selected with the consensus of all authors. Results: Several types of AI can be used in the medical field. The most common forms of AI are machine learning (ML), deep learning (DL), neural networks (NNs), natural language processing (NLP) systems, and computer vision. AI can improve various domains related to the management of urologic cancers, such as imaging, grading, and nodal staging. AI can also help identify appropriate diagnoses, treatment options, and even biomarkers. In the majority of these instances, AI is as accurate as or sometimes even superior to medical doctors. Conclusions: AI techniques have the potential to revolutionize the diagnosis, treatment, and monitoring of urologic cancers. The use of AI in urooncology care is expected to increase in the future, leading to improved patient outcomes and better overall management of these tumors.

## 1. Introduction

Medicine has changed over the decades. Due to better access to medical care, the number of patients has increased, indicating an increase in data that must be acquired and processed. Over the years, science has made numerous discoveries that can be applied to several medical issues, even in unexpected fields. It is necessary to determine how to apply these solutions to issues that seem unsuitable and even irrelevant.

The journey of Artificial Intelligence (AI) in medicine began in the 1950s and 1960s with early attempts at developing machines capable of making decisions and mimicking human conversation [1]. During the 1970s to 2000s, despite periods of reduced funding and interest, collaborations among pioneers in AI continued, leading to prototypes like the CASNET—model for glaucoma consultation. This causal-associational network included model-building, consultation, and a database, enabling personalized advice for physicians on patient management. Another milestone was the “backward chaining” AI system called MYCIN. It used patient information provided by physicians and a knowledge base of about 600 rules to suggest potential bacterial pathogens and recommend antibiotic treatments adjusted for the patient’s body weight.

The early 2000s saw a revival of interest in Machine Learning (ML) and AI with the development of the question-answering system Watson by IBM, an open-domain question-answering system. Watson harnessed the power of DeepQA technology, utilizing natural language processing and data analysis to generate probable answers from unstructured content. This breakthrough allowed for evidence-based clinical decision-making by drawing information from patients’ electronic medical records. 

With improved computer hardware and software, digitalized medicine rapidly advanced, including the use of chatbots like Siri and Alexa. Deep Learning (DL) emerged as a game-changer, allowing AI systems to classify data autonomously and process large datasets more efficiently.

Today, AI assists medical professionals in establishing diagnoses, making therapeutic decisions, and predicting the outcome. It supports every procedure that involves data processing and knowledge and is used by healthcare professionals in their everyday duties. Currently, AI can perform all these tasks with the same efficiency as skilled physicians [2]. Sometimes, it can even outperform expert clinicians [3]. 

AI is capable of a broad range of tasks, including separating cancer cells from healthy tissue, determining whether lymph node metastases have occurred, discovering biomarkers, predicting outcomes, and making therapeutic decisions [4]. In this review, we explore AI applications in urogenital system cancers, drawing from the latest research. Providing a comprehensive view of urooncology while focusing on individual cancer types, this study fosters a detailed and integrated understanding of the subject.

## 2. Materials and Methods

For this narrative review, we conducted comprehensive English-language literature research for original and review articles published until 31 December 2022 in the PUBMED/Scopus database. We searched for the following terms, alone or in combination: artificial intelligence, machine learning, deep learning, neural networks, computer-aided diagnosis, urooncology, prostate cancer, kidney cancer, testicular cancer, bladder cancer, and upper tract urothelial carcinoma. We found 249 related articles. The relevant studies were identified by evaluating the abstracts, and complete articles were obtained in cases where abstracts were unavailable. Duplicate papers were removed, and the data were screened to exclude irrelevant works. Case reports, comments, conference papers, commentaries, surveys, and animal studies were all excluded from the full-text publications. After applying the exclusion criteria, 99 full-text manuscripts were assessed for eligibility with the consensus of the authors.

## 3. Definition and Types of AI

Artificial intelligence (AI) is a broad term encompassing computer systems capable of performing tasks that traditionally require human cognition [5]. It involves programmed machines that can learn, identify patterns, and establish relationships between inputs and outputs [6]. Utilizing diverse mathematical and algorithmic methods, AI sits at the convergence of neurocomputing, statistical inference, pattern recognition, data mining, knowledge discovery, and machine learning (ML) [7]. In recent times, AI has emerged as a powerful tool, making remarkable strides in addressing numerous medical challenges.

Artificial intelligence (AI) encompasses several fields with the common goal of computationally simulating human intelligence:

1. Machine learning (ML) is one of the most important types of artificial intelligence. This technology involves prediction by identifying patterns in data using mathematical algorithms. Machine learning includes three methods: deep learning, logistic regression, and neural network architecture. ML algorithms can help automate the process of detecting and diagnosing cancer.

Deep learning (DL) predicts using multilayer neural network algorithms inspired by the neurological architecture of the brain. Deep learning (DL) can automatically extract features and assimilate and evaluate large amounts of complex data. Using large amounts of medical data and state-of-the-art computing technologies, DL can improve cancer diagnosis and treatment. DL has found widespread application in oncology research, encompassing early cancer detection, diagnosis, classification, and grading. Additionally, it has been instrumental in molecular tumor characterization, predicting patient outcomes and treatment response, facilitating personalized treatment approaches, automating radiation therapy workflows, and even aiding in the discovery of new anticancer drugs. Furthermore, DL plays a crucial role in streamlining clinical trials, revolutionizing how oncology research and patient care are conducted [8,9].

2. Neural networks (NNs) are increasingly being applied to complex ML data and include artificial neural networks (ANNs), multilayer perceptrons (MLPs), recurrent neural networks (RNNs), and convolutional neural networks (CNNs).

Artificial Neural Networks (ANNs) are computational tools inspired by the structure of the human nervous system. These networks comprise interconnected computer processors, often referred to as “neurons”, which can process data and represent knowledge through parallel computations. ANNs consist of multiple layers of neurons, including an input layer, one or more hidden layers, and an output layer. Each neuron is connected to others in the network through links, and each link possesses a numerical weight. One notable aspect of ANNs is their capacity to learn from their experiences in a training environment, making them adaptive and capable of improving their performance over time. Thanks to their analytical abilities, ANNs can compare various interactions among clinical, biological, and pathological variables and identify relationships between these variables. Researchers actively use ANNs to diagnose, treat, and predict outcomes in challenging clinical situations [9,10].

Convolutional neural networks (CNNs) are widely regarded as the most popular and effective deep learning architectures. They are particularly adept at handling large and intricate image data and extracting essential features through convolutional filters. By adjusting these filters based on learned parameters, CNNs can identify the most relevant features for specific tasks. The use of CNNs is not limited to image data; they have also been adapted to analyze non-image data, like genomic data represented in vector, matrix, or tensor formats [6,9,11]. MLPs, on the other hand, are simpler neural networks that process input data sequentially through layers, making general predictions but being susceptible to overfitting [12,13]. RNNs are designed to handle sequential data, capturing past elements in hidden “state vectors” and making predictions based on current and previous elements [12]. While some neural network models have already been approved and accepted in clinical settings, the routine clinical application of neural networks is still somewhat limited. Nevertheless, their potential for revolutionizing healthcare continues to grow, especially in fields such as cancer diagnosis and prediction [14,15].

3. NLP systems address a wide range of important clinical and research tasks. NLP is capable of processing free clinical text and generating structured output. There has been extensive focus on applying NLP techniques to identify and extract key data (information) from unstructured text so that it can be transformed into structured data that can later be analyzed and stored in a database. The steps for extracting information are as follows: named entity recognition, relationship extraction, event and temporal expression extraction, and entity merging and normalization [16].

4. Computer vision, a vital branch of artificial intelligence (AI), empowers computers and systems to derive valuable insights from digital images, videos, and visual data. By understanding and interpreting visual input, computers gain the ability to take action and provide informed recommendations. This field employs advanced deep learning technologies, particularly convolutional neural networks (CNNs), to process and analyze visual information. The primary goal of computer vision is to develop algorithms, data representations, and computer architectures that emulate human-like visual capabilities. Through computer vision, machines can “see”, observe, and comprehend the world around them, opening up new possibilities for numerous applications [16].

The discussed subfields of artificial intelligence are presented in Figure 1.

## 4. Application of AI in Urological Oncology

### 4.1. Prostate Cancer

Prostate cancer (PCa) is one of the main causes of cancer-related morbidity and mortality across the world. It is a complex and diverse disease with various diagnostic methods, including biopsy, PSA testing, and MRI. The majority of prostate cancer cases are adenocarcinomas, originating from luminal or basal epithelial cells in the peripheral regions of the prostate. Risk factors like family history, ethnicity, age, and obesity contribute to its variation across populations. Treatment options such as active surveillance, chemotherapy, radiation therapy, and surgery are tailored to individual tumor characteristics. Understanding these factors is crucial for the effective management and treatment of prostate cancer [17,18].

Given the large increase in life expectancy over the past few decades, it is reasonable to assume that the number of patients with prostate cancer will grow. Unfortunately, there are still many uncertainties surrounding the diagnosis and treatment of PCa. Consequently, it is necessary to develop new methods for managing this condition [19,20]. 

#### 4.1.1. Imaging and Diagnosis 

AI systems can automate the detection of cases in which prostate cancer is highly suspected. In a study by Cao et al. [21], the deep learning algorithm FocalNet was trained using 3T T2-weighted imaging and diffusion-weighted imaging of 553 patients who later underwent radical prostatectomy. Lesion detection sensitivity vs. the number of false-positive detections at various thresholds on suspicion scores was used to compare the PCa recognition rates of FocalNet and radiologists. For clinically important and index lesions, respectively, FocalNet performed 5.1% and 4.7% worse than radiologists. However, the differences were not statistically significant (*p* = 0.413 and *p* = 0.282, respectively) [21].

Giannini et al. proposed a computer-aided diagnosis (CAD) tool that can help manage patients suspected of having PCa and determine the target for an MRI-guided biopsy. CAD is a two-stage system. First, a map of the probability that prostate voxels will develop cancer is made. Next, to evaluate the sensitivity of the system and the quantity of false-positive (FP) regions recognized by the system, a candidate segmentation phase is carried out to highlight questionable areas. In a study by Giannini et al. [22], the area under the curve (AUC) for a cohort of 56 patients (i.e., 65 lesions) acquired during the voxel-wise phase was 0.91, and the second stage resulted in a per-patient sensitivity of 97%, with a median number of FP equal to 3, in the entire prostate sample [22]. 

CAD’s ability to detect more challenging cancers in the gland’s center may increase specificity and the radiologists’ level of experience. The objective of Gaur’s study was to compare the effectiveness of CAD to that of traditional multiparametric MRI (mpMRI) interpretation in prostate cancer identification. Index lesion sensitivities of CAD were 76% (*p* = 0.39) for the whole prostate, 77% (*p* = 0.07) for the peripheral zone, and 79% (*p* = 0.15) for the transition zone compared to those of mpMRI at 79%, 84%, and 76%, respectively [23].

In a study by Pantanowitz et al., an AI-based algorithm was created using samples of prostate core needle biopsies stained with hematoxylin and eosin. It was trained with 1,357,450 visual patches from 549 slides and tested with 2501 samples internally and with 1627 samples externally. The AUC of the algorithm was 0.997 in the internal test set and 0.991 in the external test set. The AUC for identifying Gleason pattern 5 was 0.971, and the AUC for differentiating between low-grade (Gleason score 6 or ASAP) cancer and high-grade (Gleason score 7–10) cancer was 0.941, along with 0.957 for perineural invasion. This study also provided the first instance of undetected cancer that the algorithm managed to identify [3].

Wang’s [24] prospective multi-center randomized comparative trial aimed to compare the prostate cancer (PCa) detection rate using different biopsy methods. Four hundred patients were divided into three groups: TRUS-guided 12-core standard systematic biopsy (TRUS-SB), cognitive fused mpMRI-guided 12-core biopsy (mpMRI), and artificial intelligence ultrasound of the prostate (AIUSP)-targeted biopsy. The AIUSP group showed the highest PCa detection rate (49.6%) compared to TRUS-SB (34.6%) and mpMRI (35.8%). The detection rate of clinically significant PCa (csPCa) was also highest in the AIUSP group (32.3%). The overall biopsy core positive rate was significantly lower in the TRUS-SB and mpMRI groups than in the AIUSP group. These findings suggest that AIUSP may serve as a promising alternative to systematic biopsy for PCa diagnosis in the future.

Furthermore, Stojadinović et al. demonstrated that AI can predict the risk of PCa and minimize overdiagnosis and overtreatment. They created a classification and regression tree (CART) model that, regardless of the prostate specific antigen (PSA) level, could be applied to patients referred for an abnormal PSA level, a digital rectal examination (DRE), or both in order to recognize patients with severe prostate cancer (PCa) on prostate biopsy. The CART analysis was performed using each predictor identified by the univariate logistic regression analysis. A variety of clinical utility and predictive performance aspects of risk projections were investigated. The model identified PCa in 92 (41.6%; AUC = 0.833) of 221 patients. To conclude, CART analysis prevents any major PCa from being missed while reducing unnecessary biopsies [25].

There have also been attempts to use machine learning for prostate cancer localization using transrectal ultrasound [26] and classifier ensembles using T_2_-weighted MRI alone [27].

#### 4.1.2. Gleason Grading

The ability of AI to localize, detect, and grade prostate cancer in biopsy samples can be comparable to that of prominent prostate pathology experts. Marginean et al. [28] used 698 prostate biopsy samples from 174 patients to train an AI algorithm and then tested it on 37 biopsy sections from 21 patients. AI achieved high accuracy in detecting the cancer areas, with a sensitivity of 100% and a specificity of 68%. The Gleason patterns were assigned correctly, with an intraclass correlation coefficient (ICC) of 0.96 for Gleason patterns 3 and 4, and with an ICC of 0.82 for Gleason pattern 5. Furthermore, the algorithm was comparable with pathologists in detecting cancer areas (ICC = 0.99). This discovery can undoubtedly simplify the diagnosis of prostate cancer [28]. 

Ström et al. conducted a study in which deep neural networks were trained with 6682 biopsy slides from 976 patients and tested with 1631 samples from 246 men. The assignment was to determine the occurrence, extent, and Gleason grade of malignant tissue with the help of AI. The AUC for the AI was 0.997 for differentiating benign and malignant biopsy slides. Moreover, the cancer length assigned by the reporting pathologist and predicted by the AI had a 0.96 correlation [29].

Here, the PANDA challenge ought to be mentioned. In this largest histopathology trial, 1290 developers competed to create repeatable AI algorithms for Gleason grading using 10,616 digitalized prostate samples. On external validation sets from the United States and Europe, the algorithms met expert uropathologists’ agreement levels of 0.862 (95% confidence interval (CI), 0.840–0.884) and 0.868 (95% CI, 0.835–0.900), respectively. This indicates that AI tools were capable of identifying and grading cancers with pathologist-level accuracy, achieving professional reference criteria. Furthermore, it was discovered that the algorithms missed fewer tumors than the pathologists in the US external validation set [30].

#### 4.1.3. Nodal Staging 

In a study by Hartenstein et al. [31], convolutional neural networks (CNNs) were trained to establish lymph node status in patients with PCa using only computer tomography images. Three CNNs were trained with ^68^Ga-PSMA-PET/CT imaging of 549 patients, with 2616 lymph nodes segmented. The CNNs were conducted with an AUC of 0.86. The expert clinician’s AUC was 0.81, which confirms that CNNs can adequately determine the lymphatic spread. Moreover, CNNs demonstrated the ability to “learn” since they predicted the chances of infiltration based on the anatomical regions, which positively affected their performance [31].

#### 4.1.4. Biomarkers 

AI may be valuable in analyzing and verifying potential PCa biomarkers. Green et al. attempted to determine whether the potential biomarkers Ki67 and DLX2 could be reliable indicators of PCa progression. First, they investigated the connection between tumor protein levels of Ki67 and DLX2 in transurethral resected prostatectomy samples and time to death and metastasis. Artificial neural network (ANN) analysis showed that Ki67, which was found only in 6.8% of the patients, can be predictive of reduced survival and increased probability of metastasis (*p* = 0.025), independent of the PSA level and Gleason score. Moreover, DLX2 was detected in 73% of the patients, and DLX2 was co-expressed with high Ki67 levels in 8.2% of the patients. According to ANN, DLX2 is a potential marker of increased metastasis risk. In conclusion, Ki67 and DLX2 can assist physicians in identifying patients who need to be actively monitored [32]. 

As a predictor of the presence of biomarkers, AI is faster and more objective than manual quantification. Calle et al. [33] tested the AI algorithm to identify anti-Ki67 and ERG antibodies in 648 samples. Results differed from those of manual detection by only 5% and were 100% accurate in identifying the positive tumors. Interestingly, the algorithm was also able to increase its accuracy following each round of adjustments and input from the training set [33]. 

AI may also identify promising prostate cancer biomarkers. The genetic algorithm-optimized artificial neural network (GA-ANN) was developed in order to create a diagnostic prediction model and filter potential genes obtained by meta-analysis of the openly accessible microarray data by RankProd. The expression of three genes was considered. C1QTNF3 was proven to be significantly correlated with PCa patient recurrence-free survival (RFS; *p* < 0.001, AUC = 0.57). This procedure can be used to identify other oncogenes or biomarkers in various urooncology diseases [34]. 

Proteomic analysis can also be used to discover possible biomarkers. To explore new potential proteomic signatures for prostate cancer, Kim et al. [35] created a unique method that combines targeted proteomics with computational biology. First, they identified 133 different expressed proteins in patients with PCa. Next, using synthetic peptides, they assessed these proteins in a group of 74 patients. Next, machine learning methods were used to create clinical predictive models of the diagnosis and prognosis of prostate cancer. The findings suggest that precise, noninvasive biomarkers can be found via computationally guided proteomics [35]. Furthermore, AI can predict different indicators, e.g., the 10-year cancer-specific survival (CSS) and overall survival (OS) of patients with PCa. Two gradient-boosting models using the data of patients diagnosed with PCa were trained on 7021 cases and tested on 1755 cases. The accuracy was 0.87 for the CSS and 0.98 for the OS. The ability of AI to interpret data offers clinicians and patients a new approach for predicting prostate cancer and its outcome [36].

#### 4.1.5. Treatment 

AI can determine the most appropriate treatment plan for patients. Auffenberg et al. [37] showed AI as a tool that can assist patients with PCa who have just received a diagnosis by predicting therapy choices on the basis of information from a registry of patients with comparable conditions. A prospective database of patients with PCa was built using information from 45 units of the Michigan Urological Surgery Improvement Collaborative (MUSIC). Then, a random forest machine learning model was applied, which was trained with a sample containing two-thirds of the patients and evaluated with the remaining one-third of the patients. The individualized prediction was exact (AUC = 0.81). Patients can use this online tool to obtain a better understanding of the various treatment options offered by their physicians, as well as physicians advice to seek a different therapy approach from their first choice. Indubitably, both sides can benefit from this instrument [37]. 

To predict and understand late genitourinary (GU) toxicity after radiation therapy in patients with prostate cancer, Lee et al. [38] used bioinformatics tools and machine learning techniques on genome-wide data. First, the patterns in genome-wide single-nucleotide polymorphisms (SNPs) were recognized and gathered. Next, a preconditioned random forest regression method was used to speculate on the risk on the basis of that data. The system was tested on 234 patients who had undergone radiation treatment two years earlier. The patients performed a self-assessment for four urinary symptoms using the International Prostate Symptom Score. Across the symptoms, the prediction accuracy of the method varied. It only managed to achieve a significant AUC of 0.70 for the weak stream endpoint. Nevertheless, as a result of their research, a more accurate predictive model could be created, and probable biomarkers and biological processes connected to GU toxicity could be identified [38]. 

Using megavoltage (MV) pictures for image-guided radiation therapy (IGRT) in prostate cancer patients would undoubtedly be advantageous. It eliminates the need for additional equipment and imaging doses. It additionally provides motion data with treatment beam alignment. For this purpose, Chrystall et al. [39] developed a novel real-time marker tracking system using a convolutional neural network (CNN) classifier. The CNN demonstrated high accuracy in identifying implanted prostate markers with an AUC of 0.99, a sensitivity of 98.31%, and a specificity of 99.87%. The marker tracking system achieved sub-millimeter accuracy, making it suitable for real-time applications in IGRT and providing a promising approach for accurate and efficient treatment. 

AI-based treatment planning systems are efficient and save time. In a study by Nicolae et al. [40], 41 patients who had received ^125^I low-dose-rate brachytherapy were divided into two groups randomly. The treatment of 21 patients was planned using a machine-learning-based prostate implant planning algorithm (PIPA) system, while the treatment of the remaining patients was planned using the conventional method. After the radiation oncologist modified the plan, the first evaluation was carried out by determining the dice coefficient of the prostate V150% isodose volume between PIPA and the standard method. Additional comparisons between groups focused on dosimetric results at preimplant and Day 30, as well as the amount of planning time. Results indicated that the plans of more patients’ in the PIPA group did not require modification, and compared to the conventional technique (43.13 ± 58.70 min), the planning time for PIPA was significantly shorter (2.38 ± 0.96 min). In addition, no discernible differences between these two groups were discovered [40].

Deng et al. showed a similar strategy. Docetaxel, the medication used to treat metastatic castration-resistant prostate cancer, is only effective in 20% of patients. Al can accurately divide patients into docetaxel-tolerant and docetaxel-intolerant groups, which can help select adequate treatment and avoid early therapeutic failure [41]. 

Table 1 summarizes the most important studies examining AI applications in prostate cancer.

### 4.2. Kidney Cancer

Renal cell carcinoma is a diverse group of cancers with various genetic and molecular alterations, including clear-cell, papillary, and chromophobe subtypes. Established risk factors include tobacco smoking, hypertension, and obesity. Renal carcinoma can often remain clinically silent until reaching an advanced stage. Classic symptoms, such as pain, haematuria, and flank mass, occur in only a small percentage of cases. Routine imaging has become instrumental in identifying renal cell carcinoma incidentally. The most crucial staging technique is computed tomography (CT) of the abdomen. Survival rates are more favorable when tumors are confined to the kidney. However, renal carcinoma is notoriously resistant to chemotherapy, making radical nephrectomy the standard treatment [43,44].

Machine learning models and deep learning algorithms are used to diagnose renal tumors, differentiate benign and malignant renal tumors, and differentiate renal cell carcinoma (RCC) types. Table 2 summarizes the research looking at applications of AI in kidney cancer.

#### 4.2.1. Prediction and Detection of Kidney Cancer

The incidence of kidney cancer is increasing every year. In 2020, the number of cases of RCC registered in the United States was 44,400. Researchers predict that, by 2050, the number of RCC cases will rise to 55,400. Consequently, several studies are attempting to counter this trend with new ways to predict and detect kidney cancer. Santoni et al. [45] attempted to implement an artificial neural network to predict new cases in the population. They used data such as population index, obesity, smoking prevalence, uncontrolled hypertension, and life expectancy data in the United States. The study involved collecting statistics on the US population and assessing how various factors affect the incidence of kidney cancer. MATLAB R2018 (MathWorks) software was used to implement an artificial neural network. As per the results, hypertension prevention has the greatest impact on reducing the incidence of kidney cancer. The study estimated that, by preventing hypertension, it will be possible to reduce the incidence of kidney cancer by 575 cases per year by 2030. Other factors had a more limited impact [45]. A group of researchers led by Houshyar [46] retrospectively analyzed CT images of 319 patients. They created two separate CNNs. The first CNN focused on localizing the bounding cube of the right and left kidney hemispheres, while the second CNN focused on segmenting the renal parenchyma and tumors within each cube. The performance of the CNNs was evaluated in a cohort of 269 patients. The median Sorensen-Dice coefficients for kidney and tumor segmentation were 0.970 and 0.816, respectively, indicating accurate delineation. Moreover, the Pearson correlation coefficients between the CNN-generated and human-annotated estimates of kidney and tumor volumes were 0.998 and 0.993, respectively (*p* < 0.001), confirming the reliability of the CNN approach.

These preliminary findings demonstrate the potential of automated deep learning AI techniques for rapid and precise segmentation of kidneys and renal tumors on single-phase contrast-enhanced CT scans. Additionally, CNNs can accurately calculate tumor and kidney volumes, offering valuable assistance in clinical practice [46].

#### 4.2.2. Differentiation of Benign and Malignant Renal Tumors

Distinguishing between benign and malignant tumors is crucial, as benign tumors, like adiposarcoma (AML) and oncocytoma, are sometimes misclassified as RCC. This differentiation is essential to avoiding unnecessary medical procedures. Recent advancements in machine learning and deep learning using radiomics have shown promise in accurately differentiating these tumors [57]. Erdim et al. [47] conducted a study where they analyzed CT images of benign and malignant tumors, extracting texture features to create a predictive model with machine learning algorithms. The random forest algorithm, utilizing five selected contrast-enhanced CT texture features, demonstrated the best predictive performance with an accuracy of 90.5% and an area under the curve (AUC) of 0.915.

Another study by Uhlig et al. [48] compared the diagnostic accuracy of two experienced radiologists with the random forest algorithm in evaluating renal masses from CT imaging. The results showed that the random forest algorithm outperformed the radiologists in correctly identifying and classifying renal masses.

These findings highlight the potential of machine learning algorithms in accurately differentiating between benign and malignant renal tumors, providing valuable assistance in clinical decision-making, and optimizing patient care.

#### 4.2.3. Differentiation of RCC Types

Radiomics analysis has led to the development of machine learning and deep learning models to distinguish between the five types of RCC, including oncocytoma, AML, clear cell RCC, papillary RCC, and chromophobe RCC. CT texture analysis is utilized by ML and DL algorithms to differentiate various renal masses [43,44,45]. ML and DL algorithms can predict nuclear class and identify specific genetic mutations, which affect the prediction of prognosis, recurrence, and survival outcomes [58].

In their study, Kocak et al. [59] used 275 textural features from CT images to predict and identify the nuclear class of ccRCC. The machine learning models performed well in differentiating different forms of renal cell carcinoma (RCC), but their identification of the three basic types was poor. SVM showed the highest predictive value for nuclear grades in ccRCC cases (85.1%). Cortico-subcortical CT images provided more texture parameters than nonenhanced images.

Uhm et al. [49] Uhm et al. developed a dataset of 1035 CT images from 308 patients containing five major types of renal tumors. They compared the diagnostic performance of their deep learning model with that of six radiologists. The AI outperformed radiologists in diagnosing most types of RCC and benign tumors, showing significantly better results in diagnosing oncocytoma and liposarcoma AML tumors. However, the results were similar for the diagnosis of clear-cell RCC.

Nikpanah et al. [50] conducted a retrospective study involving 74 patients with 243 renal masses to assess a deep convolutional neural network’s diagnostic efficacy in distinguishing clear-cell renal cell carcinoma from renal oncocytoma. MR imaging was performed before pathologic confirmation, and a deep neural network (AlexNet) was fine-tuned for this task. The AI system achieved an overall accuracy of 91% and an area under the curve of 0.9 in distinguishing ccRCC from oncocytoma using fivefold cross-validation. Utilizing features extracted from 20,000 CT images, Pedersen et al. [60] created convolutional neural networks that exhibited a remarkable 93.3% accuracy and a specificity of 93.5% in effectively differentiating oncocytoma from RCC.

A group of researchers led by Tabibu et al. [51] conducted a study that explored the application of a deep learning method to identify and classify different RCC types, achieving an impressive classification accuracy of 94.07%. They introduced a novel support vector machine-based approach to enhance model performance in multiclass classification (pan-RCC), resulting in a remarkable 93% accuracy in cancer detection. Furthermore, the researchers utilized morphological features extracted from tumor regions identified by CNNs to predict survival outcomes for patients with the prevalent clear-cell RCC.

#### 4.2.4. Differentiation Grade of Clear Renal Cell Carcinoma (Fuhrman Grade)

The Fuhrman grading system [61] evaluates nuclear size, shape, and nucleolar prominence, categorizing tumors into four nuclear grades (1–4) based on increasing testicular size, irregularity, and prominence. This grading system is a robust predictor of distant metastasis after nephrectomy. Metastasis rates correlate with nuclear grade, with grade 1 tumors exhibiting significantly lower rates compared to grades 2 to 4. Survival outcomes are also stratified into three categories: grade 1, grade 4, and grades 2 and 3.

Ding et al. [52] illustrated enhanced precision in staging the classification of clear cell renal cell carcinoma by preoperatively distinguishing between high-grade (Fuhrman III–IV) and low-grade (Fuhrman I–II) tumors. Their algorithm integration encompassed six key non-textural features: pseudocapsule, round mass, maximum tumor diameter (Diametermax), intracellular artery (Arteritumor), tumor enhancement value (TEV), and relative TEV (rTEV), alongside texture features. Extracted from CT images of the segment with the largest renal mass area in the corticospinal and nephrographic phases, these texture features underwent selection through the least absolute shrinkage and selection operator (LASSO) to calculate a texture score for each patient. In their approach, a logistic regression model utilizing three iterations—model 1 with all non-texture features, model 2 with all non-texture features and texture score, and model 3 with only texture score—distinguished high-grade ccRCC from low-grade ccRCC during nephrectomy. These models exhibited strong discrimination in the training cohort, yielding area under the receiver operating characteristic curve (AUC) values of 0.826, 0.878, and 0.843 for models 1, 2, and 3, respectively. Notably, a significant difference in AUC was observed between model 1 and model 2.

Tian et al. [62] developed a CAD system for Fuhrman classification in ccRCC using 395 whole-mount images. Their model, incorporating 26 features, predicted tumor grade with 84.6% sensitivity and 81.3% specificity. Their results were significantly related to overall survival. Holdbrook et al. [63] concentrated on nuclear pleomorphic patterns, creating a binary CAD system for renal cell carcinoma grading. This system demonstrated high accuracy (F-score = 0.78–0.83) and predicted survival with similar precision as an established scoring system based on multigene testing.

A different approach was shown by Wen-Zi [64]. The study utilized deep learning algorithms to predict the pathological staging and grading of tumors in 878 patients based on preoperative clinical variables. The proposed models, including BiLSTM, CNN-BiLSTM, and CNN-BiGRU, achieved impressive accuracy in predicting tumor pathological staging, with AUC values of 0.933, 0.947, and 0.948, respectively. For tumor pathological grading, the models yielded AUC values of 0.754, 0.720, and 0.770, respectively.

#### 4.2.5. Genetic Mutation

Li et al. [65] harnessed gene expression, machine learning (utilizing random forest variable hunting), and Cox regression analysis to construct a risk score model based on 15 genes. This model aimed to predict the survival of ccRCC patients in the Cancer Genome Atlas dataset (N = 533). Remarkably, the higher-risk group demonstrated significantly worse prognosis and survival compared to the lower-risk group. A similar pattern was observed in recurrence-free survival. Interestingly, the risk scores were not correlated with patient characteristics such as gender or age but were linked to hemoglobin levels and tumor characteristics like size and grade. Notably, radiation therapy had no influence on the predictive value of the risk score. Multivariate Cox regression underlined the importance of the risk score as an indicator of prognosis in ccRCC. Ultimately, this risk score model, driven by the expression of 15 selected genes, exhibited the ability to predict the survival of ccRCC patients.

In their research, Kocak et al. [53] employed both an artificial neural network (ANN) algorithm and a random forest (RF) algorithm to detect PBRM1 mutations through CT texture analysis. The ANN demonstrated accurate identification of 88% of ccRCCs with a PBRM1 mutation status, while the RF algorithm showed even higher performance, correctly classifying 95.0% of ccRCCs with a PBRM1 mutation status.

Machine learning has demonstrated high accuracy in distinguishing CD117 (c-KIT) oncocytomas from the chromophobe subtype of renal cell carcinoma using the peak early enhancement rate (PEER), with a 95% accuracy for tumor type classification (100% sensitivity and 89% specificity) [66].

In their study, Tian et al. [54] investigated mRNA expression profiles in the GSE53757 dataset and their relation to the clinical prognosis of renal cell carcinoma. They developed a seven-gene independent prognostic model that showed significant correlations with the prognosis of renal cell carcinoma. The researchers carefully selected renal cancer differentiation genes from the dataset and performed functional enrichment analysis, revealing enriched biological functions related to catabolic processes of small molecules, T-cell activation, and other aspects.

Tian et al. [54] employed RF and SVM models to refine their prognostic model, identifying seven hub mRNAs through Cox LASSO analysis as robust predictors of kidney cancer prognosis. Subsequent measurement of these seven genes in kidney cancer and normal tissue pairs revealed higher expression of RNASET2 and FXYD5 in cancer tissue, while NAT8 exhibited relatively lower expression. However, no significant differences were observed in the expression of EZH2, KLF18A, CDCA7, or WNT7B between tumor tissue and adjacent tissue.

Tian et al. [54] utilized bioinformatics technology to integrate genomic data and identify differentially expressed genes (DEGs) associated with renal cell carcinoma prognosis. Their developed prognostic mRNA model outperformed single mRNA models, effectively distinguishing patients at high risk of recurrence from those at low risk. The model’s prognostic performance remained independent of age and stage. The study suggests that a nomogram combining seven gene signatures can accurately depict the risk level and overall survival based on the patient’s clinical stage, age, and other factors.

#### 4.2.6. Treatment of Kidney Cancer

In 2003, Kattan et al. [67] published experiments with AI. The authors compared AI and Cox regression to predict disease recurrence after surgery. In their experiments, Cox regression models showed better performance. For kidney cancer, they used ANN and Cox regression to predict behavior (median accuracy of 71% for ANN and 75% for Cox). Khene and a group of collaborating researchers studied the response of patients to nivolumab therapy. Nivolumab serves as an effective immunotherapy with checkpoint inhibition in mRCC [68]. They showed that pretreatment imaging radiomics could accurately identify those responding to nivolumab. The model could achieve an accuracy of more than 90% in predicting treatment response [69]. Buchner et al. [55] examined the potential of AI in predicting the outcome of patients with metastatic renal cell carcinoma who were about to start systemic therapy. The AI model was trained using data from 175 patients who had undergone radical or partial nephrectomy of the primary tumor prior to commencing systemic therapy. The main objective was to predict overall survival at the 3-year mark based on parameters available at the initiation of first-line therapy. AI was able to achieve an accuracy of 95%. 

The Barkan et al. [56] study aimed to assess the capabilities of emerging AI technologies in predicting three- and five-year overall survival (OS) for patients with advanced metastatic renal cell carcinoma (mRCC) undergoing their first-line systemic treatment. The retrospective analysis included 322 Italian patients treated between 2004 and 2019. An ensemble of three AI predictive models was developed, outperforming existing prognostic systems and providing better clinical support for decision-making. The model achieved high accuracy and specificity, with AUC values of 0.786 and 0.771 for 3-year OS and 5-year OS, respectively. The AI models demonstrate promising potential for enhancing patient management in mRCC treatment, but larger studies are needed to validate their effectiveness further.

The objective of Le et al.’s [70] study was to develop and validate predictive models using machine learning algorithms for patients with bone metastases (BM) from clear cell renal cell carcinoma (ccRCC) and to identify suitable models for clinical decision-making. The researchers obtained data from the Surveillance, Epidemiology, and End Results (SEER) database for 1490 ccRCC-BM patients and collected clinicopathological information for 42 patients at their hospital. Four ML algorithms (extreme gradient boosting, logistic regression, random forest, and Naive Bayes model) were employed to predict overall survival (OS) in ccRCC-BM patients. The patients were divided into training and validation cohorts for evaluation. The models performed well in predicting 1-year and 3-year OS, suggesting that ML can be a valuable tool in clinical decision-making for ccRCC-BM patients.

AI can also be used to assess recurrence risk following surgical resection of RCC. Khene et al.’s [71] study explored the effectiveness of machine learning models, including Random Survival Forests, Survival Support Vector Machines, and Extreme Gradient Boosting, in predicting recurrence after surgical resection of nonmetastatic renal cell carcinoma. Conducted across 21 French medical centers with over 4000 patients, the ML models outperformed traditional prognostic models in predicting disease-free survival. ML models demonstrated higher concordance index values, good calibration, and superior net benefit in decision curve analysis.

From this research, it can be concluded that artificial intelligence methods are future-proof. Tests using new technologies can, taking into account predisposing behavioral factors, not only detect the disease at an early stage but also help actively control predisposed individuals. Deep learning methods reduce the waiting time for histopathology results and can help differentiate types of kidney cancer. With the help of AI, it is possible to assess the response to gene therapy and, taking into account genetic mutations, to evaluate prognosis and survival time. 

### 4.3. Bladder Cancer

Bladder cancer is a prevalent malignancy affecting both men and women, with the most common type being transitional cell carcinoma arising from urothelial cells in the bladder. Its primary symptoms include hematuria and lower urinary tract issues. Thanks to advanced imaging and diagnostic tools, bladder cancers are now more likely to be detected in their early stages. About 75% of cases are non-muscle-invasive and treated with transurethral resection of the tumor, while the remaining have invaded deeper layers or formed metastases, necessitating radical cystectomy [72,73,74]. 

#### 4.3.1. Diagnosis 

Cystoscopy is considered the gold standard for diagnosing and monitoring non-muscle-invasive bladder cancer (NMIBC). However, this procedure is not a perfect test. In addition to being user-dependent, white-light cystoscopy can also have various limitations; small tumors, carcinoma in situ, and other nonobvious lesions in certain anatomic locations can be easily missed. Due to the increasing number of different upgrades, such as blue-light cystoscopy (BLC) and artificial intelligence, the procedure is becoming more accurate [75]. 

Various researchers of AI methods have evaluated the effectiveness of machine learning in overcoming human mistakes and ensuring that diseases are not missed. Some of the main algorithms used to improve cystoscopic diagnosis in addition to survival and prognosis prediction in bladder cancer are convolutional neural networks (CNNs), multilayer perceptrons (MLPs), support vector machines (SVMs), and genetic algorithms (GAs). Ikeda et al. [76] aimed to support the cystoscopic diagnosis of bladder cancer using a convolutional neural network. They created a CNN-based tumor classification. The trained classifier dataset consisted of 2102 cystoscopic images (1671 images of normal tissue and 431 images of tumor lesions). Its effectiveness was evaluated using test data (87 tumor images and 335 normal images). In the result, 78 images were true positives, 315 were true negatives, 20 were false positives, and nine were false negatives (i.e., sensitivity was 89.7% and specificity was 94.0%). Eminaga et al. [77] applied the same kind of AI technology, creating various deep CNN models and assessing them using the F1-score. The greatest F1-score, 99.52%, was obtained for the XCeption-based model. Other models that could identify all cystoscopic images with bladder were based on ResNet50 and the harmonic series concept, which achieved F1-scores of 99.48% and 99.45%, respectively. 

The multilayer perceptron (MLP)-based method, presented by Lorencin et al. [78], uses image resizing and a Laplacian edge detector for the preprocessing of input images. The method provides an alternative strategy for diagnosing bladder cancer. AI was trained with the data of 1997 and 986 images with and without bladder cancer, respectively. Results were encouraging, with an AUC value of up to 0.99. Hashemi et al. [79] applied the local binary pattern (LBP) method to extract the features in bladder cystoscopy images. Then, they used the MLP neural network to train and evaluate the classifier for images from a bladder cystoscopy. In order to enhance the performance of this method, the researchers applied an adaptive learning rate and a genetic algorithm. The simulated results revealed a significant 7% reduction in error and improved convergence speed compared to other competing methods. The findings underscore the immense potential of deep learning for accurately diagnosing cystoscopic images.

#### 4.3.2. Metastasis Detection

A particularly important thing in determining the patient’s condition is to confirm the presence of metastases in the lymph nodes. In Gresser et al.’s [80] study, a radiomics signature was developed using machine learning to detect lymph node metastases in bladder cancer patients who underwent radical cystectomy with lymphadenectomy. Out of 1354 patients screened, 391 with pathological nodal staging were included and divided into training and test cohorts. Radiomics features were extracted from each lymph node, and an ML model was trained using histopathology labels. Manual and automated lymph node segmentations were compared to radiologist assessments for detecting metastases. The results showed that the radiomics-based analysis using manual lymph node segmentation achieved an AUC of 0.80, while the fully automated approach achieved an AUC of 0.70. Combining the manually segmented radiomic signature with radiologist assessment improved the AUC to 0.81.

In Wu et al.’s [81] study, researchers developed a lymph node metastases diagnostic model (LNMDM) using whole slide images and assessed the clinical impact of an artificial intelligence-assisted (AI) workflow. The LNMDM was developed using data from 998 bladder cancer patients who underwent radical cystectomy and pelvic lymph node dissection. The model demonstrated high diagnostic sensitivity, with an area under the curve (AUC) ranging from 0.978 to 0.998 in five internal validation sets. Comparisons between the LNMDM and pathologists showed that the model outperformed both junior and senior pathologists in detecting lymph node metastases. AI assistance improved sensitivity for pathologists, enhancing diagnostic accuracy. Notably, the model identified tumor micrometastases that had been missed by pathologists in some cases.

Another study [82] aimed to develop and validate a machine-learning-based approach using [^18^F]FDG PET/CT criteria to accurately identify pelvic lymph node involvement in patients with muscle-invasive bladder cancer (MIBC). The study consisted of 173 patients. The developed machine-learning-based combination of criteria, which included features from pelvic lymph nodes and the primary bladder tumor, showed comparable diagnostic performance (AUC = 0.59) to the consensus of experts (AUC = 0.64) in the validation set. The interrater agreement was also good (Κ = 0.66) for both the machine-learning approach and the experts.

#### 4.3.3. Prediction and Prognosis 

Machine learning algorithms can be used to improve more than just cystoscopy. The other aspects of bladder urooncology where artificial intelligence may be employed are the prediction and prognosis of mortality, postcystectomy recurrence and survival, and therapy response. Deep learning systems (DLSs) are also being developed for clinical cytology in order to detect the malignant potential of urothelial carcinoma cells. 

Wang et al. [83] employed a least squares support vector machine (SVM) to predict the 5-year overall and cancer-specific mortality of patients who underwent radical cystectomy. The model achieved an accuracy of over 75% in this prognostic prediction. [4]. To predict the prognosis over the next 5 years using various combinations of image, clinical, and spatial features, Gavriel et al. [84] proposed an ensemble system that consists of ML-based algorithms. The method demonstrated a 71.4% accuracy in correctly identifying patients who experienced unfavorable outcomes and succumbed to muscle-invasive bladder cancer (MIBC) within a 5-year timeframe. This value is impressive considering that it is significantly higher than the 28.6% of the TNM staging system, the current clinical gold standard. 

#### 4.3.4. Disease Progression and Chemotherapy Efficacy

ML-based algorithms and models have been used in several articles to identify genes that can potentially foretell the recurrence or future progression of disease. Slides from patients with MIBC were marked with immunofluorescence (IF) and then applied to measure the tumor buds. In this way, the efficacy of neoadjuvant chemotherapy was evaluated, and the patients who did not respond to the therapy were identified, with the aim being to stop the treatment midway in such patients to avoid the harmful effects of chemotherapy [4,85,86,87]. Nojima et al. [88] developed a DLS to detect high-grade urothelial carcinoma (HGUC) cells in urine cytology specimens using a pretrained VGG16 model. The DLS demonstrated exceptional performance when trained on high-power field images of both malignant and benign cases. It accurately diagnosed invasive UC lesions with an AUC of 0.8628 and an F1-score of 0.8239. Moreover, it successfully identified high-grade UC lesions with an AUC of 0.8661 and an F1-score of 0.8218.

The results indicated that the DLS exhibited the potential to more accurately determine the malignant potential of tumors compared to classical cytology. Considering this possibility, along with improvements in prognosis and prediction, urologists may be better able to develop therapeutic strategies that will ultimately benefit patients.

Table 3 summarizes the studies examining the use of AI in bladder cancer.

### 4.4. Upper Tract Urothelial Carcinoma (UTUC)

Upper tract urothelial cancer (UTUC), a specific type of urothelial cancer, occurs in the ureter and renal pelvis. It is a relatively rare cancer, accounting for 5–10% of all urothelial carcinomas. A significant number of UTUC tumors are invasive at the time of diagnosis, and the 5-year cancer-specific survival rates for advanced stages are low. The standard treatment for high-risk UTUC is radical nephroureterectomy (RNU) with bladder-cuff removal. However, kidney-sparing surgery (KSS) is gaining popularity to preserve renal function, akin to managing parenchymal renal cancer. Neoadjuvant chemotherapy shows potential benefits for high-risk UTUC patients [89].

Currently, the gold standard for UTUC diagnosis and conservative treatment is flexible ureteroscopy (URS). URS allows for a thorough examination of the urinary system tissue, identification of tumors, assessment of their size, and biopsy of suspicious lesions [90]. The procedure is performed with the assistance of an endoscopic camera to provide visual guidance [91].

Similar to other endoscopic procedures, in this procedure, too, artificial intelligence can be used to enhance the outcomes.

Primary research on this subject, presented by Lazo et al. [92], provided an automated method based on convolutional neural networks to produce an accurate segmentation of the hollow lumen. The described method included an ensemble of four parallel CNNs (U-Net-based, Mask-RCNN, and two modifications of the former ones) to process both single-frame and multi-frame data simultaneously. Using a unique dataset of 11 recordings (2673 frames) that were gathered from six patients and manually annotated, they evaluated the proposed method, which outperformed earlier state-of-the-art techniques with an F1-score of 80%. Although the results demonstrated that the ensemble model may successfully enhance hollow lumen segmentation in ureteroscopic images, the development of the submitted method might further the UTUC finding, particularly if it works effectively even when there is limited visibility, occasional bleeding, or specular reflections.

### 4.5. Testicular Tumors

Testicular cancer is a prevalent solid malignancy affecting young adult men, and its occurrence has been on the rise globally. Testicular cancer accounts for about 1% of newly diagnosed cancers in men globally and is most common in men aged 14 to 44 in Western countries. Cryptorchidism, a birth defect where one or both testicles are not in the scrotum, is a significant risk factor for testicular cancer, increasing the risk nearly fivefold. Other potential risk factors include hypospadias and a low sperm count. The precursor lesion to malignant testicular germ cell tumors is germ cell neoplasia in situ (GCNIS). Germ cell tumors (GCT) constitute the majority of testicular cancers and are divided into seminoma and nonseminoma subtypes. Seminomas are homogeneous tumors of embryonic germ cells, while nonseminomas comprise various histologic subtypes. Advances in testicular cancer management have led to remarkable success, with a cure rate exceeding 70% for the first metastatic solid tumor [93,94].

Lymphovascular invasion (LVI) holds significant prognostic value, particularly in stage 1 non-seminomatous tumors and germ cell tumors of the testis. LVI refers to tumors found within endothelium-lined lymphatic or vascular channels. To automate the identification of suspected LVI areas in digital whole-slide images of testicular tumors, Ghosh et al. [95] developed an artificial intelligence algorithm using deep learning. They trained the algorithm with 184 histology slides stained with hematoxylin and eosin (H&E) from 19 patients with testicular germ cell tumors. The algorithm successfully identified areas of potential LVI in a validation set of 118 whole-slide images from 10 patients, with a precision of 0.68 for suitable areas and 0.56 for definite LVI areas. This proof-of-concept study demonstrates the feasibility of an artificial intelligence tool that aids reporting pathologists in highlighting areas for potential LVI assessment [95].

Distant metastasis in testicular cancer patients, beyond non-regional lymph nodes and lungs, presents a significant concern. Ding et al. addressed this issue by developing a machine learning (ML) algorithm to predict the risk of patients with germ cell testicular cancer (GCTC) progressing to the M1b stage, enabling early intervention. The predictive model was constructed using data from 4323 GCTC patients obtained from the Surveillance, Epidemiology, and End Results (SEER) database. Six ML algorithms were utilized to build the model, demonstrating high accuracy, stability, and computational efficiency. These promising results have valuable implications for clinical decision-making and provide a potential tool for timely interventions in GCTC patients [96].

Linder et al. [97] developed a deep learning approach to identify and count tumor-infiltrating lymphocytes (TILs) in primary testicular cancer patients. They analyzed H&E-stained whole slides from 113 patients and found a significant association between low TIL numbers and disease recurrence. A higher TIL density was correlated with a lower clinical tumor stage, seminoma histology, and absence of lymphovascular invasion at presentation.

Baessler et al. [98] used ML-based CT radiomics to distinguish between malignant and benign lymph nodes in patients with retroperitoneal LN metastases from NSTGCT, aiming to reduce overtreatment in young patients. The model achieved an accuracy of 81%, a sensitivity of 88%, and a specificity of 72%.

Another study [99] compared custom-designed and commercial ANNs for staging testicular cancer using pathological parameters. The custom ANN outperformed the commercial ANN (92% vs. 80% accuracy), highlighting the importance of individual network refinement by investigators, which currently limits widespread commercial adoption of these methods.

## 5. Conclusions

Less than 20 years after the dawn of computing, AI has been applied to clinical decision-making. However, only recently, with the development of machine learning, has it been integrated into clinical practice. AI methods for analyzing big data cohorts seem more precise and exploratory than conventional regression statistics. Additionally, they provide specific health behavior predictions. Each artificial intelligence method has features that make it effective for various tasks. Understanding the basics of AI approaches and their potential, especially the flexibility of certain systems, will enable these innovative methods to be developed further and play a significant role in urologists’ work with patients with cancer.

## Figures and Tables

**Figure 1 cancers-15-04282-f001:**
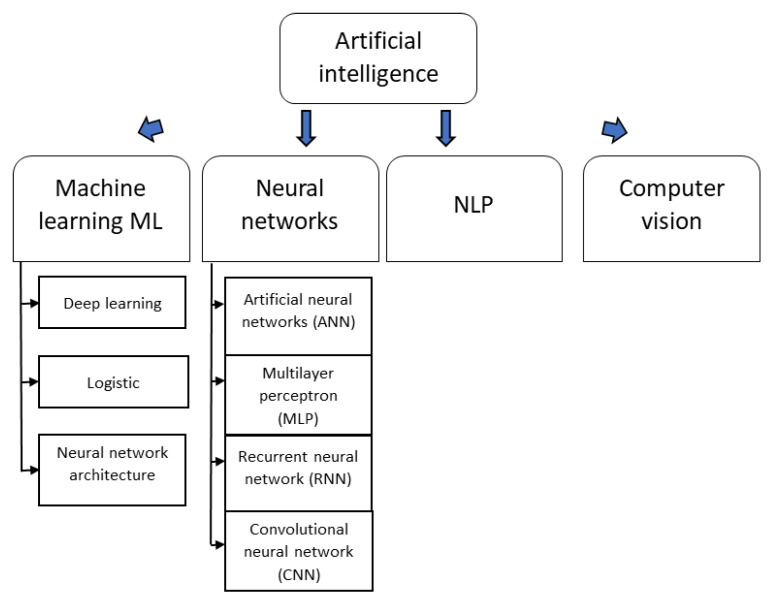
Subfields of Artificial Intelligence.

**Table 1 cancers-15-04282-t001:** Studies looking at applications of AI in prostate cancer.

Study	Objective	Algorithm/Method	Study Design	Results
Cao et al. [21]	Detection of prostate cancer using 3 T multiparametric magnetic resonance imaging	Deep learning algorithm	development cohort: 427 patients evaluation cohort: 126 patients	Detection sensitivity: 5.1% and 4.7% below the radiologists for clinically significant and index lesions, respectively
Giannini et al. [22]	Setting of the MRI-guided biopsy target	Computer-aided diagnosis	56 patients 65 lesions	Accuracy—97%
Gaur et al. [23]	Detection of prostate cancer using mpMRI	Computer-aided diagnosis	144 case patients 72 control patients	Improved patient-level specificity (72%) compared to mpMRI-alone (45%)
Wildeboer et al. [42]	Detection of prostate cancer using B-mode, shear-wave elastography, and contrast-enhanced ultrasound radiomics	Machine learning	48 patients	AUC-ROC of 0.75 for detecting PCa and 0.9 for detecting Gleason score greater than 3 + 4
Viswanath et al. [27]	Detection of peripheral zone prostate tumors using T_2_-weighted MRI	Computer-aided diagnosis	85 prostate cancer datasets acquired from across 3 different institutions (1 for discovery, 2 for independent validation)	AUC of 0.744 for detecting PCa
Marginean et al. [28]	Standardization of Gleason grading in prostate biopsies	Machine learning and convolutional neural networks	698 prostate biopsy sections from 174 patients for training 37 biopsy sections from 21 patients for test	Sensitivity in detecting cancer (100%) and identifying the correct Gleason pattern (80–91%) depending on the Gleason pattern, and specificity (68–98%) depending on the Gleason pattern.
Ström et al. [29]	Detection and grading of prostate cancers in prostate biopsies	Deep neural networks	6953 prostate biopsy samples from 1063 patients for training 1943 biopsy samples from 391 patients for evaluations	AUC-ROC of 0.997 in distinguishing the malignancy, comparable performance to expert pathologists in assigning Gleason grades.
Bulten et al. [30]	Detection and grading of prostate cancers in prostate biopsies	Multiparametric algorithms	10,616 digitized prostate biopsies	Agreements of 0.862 and 0.868 with expert uropathologists
Hartenstein et al. [31]	Prostate cancer nodal staging using CT imaging	Convolutional neural networks (CNNs)	2616 lymph node samples from 546 patients were segmented	AUC of 0.95 and 0.86 compared to an AUC of 0.81 for experienced radiologists
Green et al. [32]	Identification and validation new biomarkers	Artificial neural network (ANN)	192 tissue microarrays (TMA) constructed from transurethral resected prostatectomy histology samples	High Ki67 is predictive of reduced survival and increased risk of metastasis, independent of PSA and Gleason score. DLX2 shows increased metastasis risk and co-expression with a high Ki67 score
Calle et al. [33]	Automation analysis of biomarkers	Deep learning algorithm	648 samples of tissue microarrays (TMA)	5% variance compared to manually generated results; 100% accuracy in identifying positive tumors
Hou et al. [34]	Identification and validation of new biomarkers	Genetic algorithm-optimized artificial neural network (GA-ANN)	Meta-analysis using RankProd from microarray data	AUC of 0.953 for diagnostic accuracy and AUC of 0.808 for prognostic capability
Auffenberg et al. [37]	Development of a web-based system to provide newly diagnosed men with predicted treatment decisions	Random forest ML model	Registry data from 45 MUSIC urology practices from 2015 to 2017	AUC of 0.81 for personalized prediction
Lee et al. [38]	Prediction of late GU toxicity after prostate radiation therapy	Preconditioned random forest regression method	324 patients at 2 years post-radiation therapy	Accuracy—70%

**Table 2 cancers-15-04282-t002:** Studies looking at applications of AI in kidney cancer.

Study	Objective	Algorithm/Method	Study Design	Results
Santoni et al. [45]	Prediction of new cases of RCC	ANN	Statistics on US population numbers	24.7% increase in new RCC cases, rising from 44,400 in 2020 to 55,400 in 2050
Houshyar et al. [46]	Development of a surgical planning aid	CNN	CT images of 319 patients	Median Dice coefficients for kidney and tumor segmentation were 0.970 and 0.816, respectively.
Erdim et al. [47]	Distinguishing between benign and malignant solid renal masses	ML	21 patients with benign renal masses 63 patients with malignant renal masses 271 texture features extracted from CT images	Best predictive performance with an accuracy of 90.5% and an AUC of 0.915
Uhlig et al. [48]	Distinguishing between benign and malignant clinical T1 renal masses	Random forest algorithm	48 patients	AUC of 0.83 compared to radiologists’ 0.68, sensitivity 0.88 vs. 0.80, *p* = 0.045, specificity 0.67 vs. 0.50, *p* = 0.083
Uhm et al. [49]	Differentiation of RCC types	DL	Dataset of 1035 CT images from 308 patients containing five major types of renal tumors	AUC of 0.855, comparable diagnostic performance to that of radiologists
Nikpanah et al. [50]	Distinguishing clear cell renal cell carcinoma from renal oncocytoma	Deep neural network (AlexNet)	74 patients with 243 renal masses	Overall accuracy of 91% and an AUC of 0.9
Tabibu et al. [51]	Differentiation of RCC types	CNN	Histopathological images from patients with RCC subtypes	Accuracy of 93.39% for distinguishing clear cell and chromophobe RCC from normal tissue; accuracy of 94.07% for distinguishing clear cell, chromophobe, and papillary RCC subtypes
Ding et al. [52]	Differentiation grade of ccRCC	CT-based radiomic models	14 patients with ccRCC who underwent partial or radical nephrectomy	AUC of 0.826, 0.878, and 0.843 for models 1, 2, and 3, respectively
Kocak et al. [53]	Detection PBRM1 mutations through CT texture analysis	ANN and RF	45 patients with clear-cell RCC, among whom 16 had the PBRM1 mutation	ANN algorithm’s AUC of 0.925, RF algorithm’s AUC of 0.987
Tian et al. [54]	Screening for kidney cancer prognosis biomarkers	RF	Kidney cancer RNA sequencing data from the Gene Expression Omnibus (GEO) database	In tumor tissue, RNASET2 and FXYD5 were found to be highly expressed, while NAT8 was observed to be lowly expressed at both the protein and transcription levels
Buchner et al. [55]	Prediction of the metastatic RCC outcome	ANN	175 patient records with available follow-up data for a median of 36 months	95% overall accuracy, outperforming logistic regression models (78% accuracy)
Barkan et al. [56]	Predicting OS for mRCC patients	ML	322 patients	AUC of 0.786 for three-year OS and 0.771 for five-year OS

**Table 3 cancers-15-04282-t003:** Studies looking at applications of AI in bladder cancer.

Study	Objective	Algorithm/Method	Study Design	Results
Ikeda et al. [76]	Improvement of the quality of bladder cancer diagnosis by supporting cystoscopic diagnosis using AI	Convolutional neural network (CNN)	1671 cystoscopic images of normal tissue 431 cystoscopic images of tumor lesions	AUC-ROC of 0.98 in distinguishing normal and tumor tissue
Eminaga et al. [77]	Exploration of the potential of AI for the diagnostic classification of cystoscopic images	Convolutional neural network (CNN)	18,681 cystoscopic images from 479 patients	CNN achieved F1 scores of 99.52%, 99.48%, and 99.45%
Lorencin et al. [78]	Investigation of the MLP implementation possibility for the detection of urinary bladder cancer	Multi-Layer Perceptron (MLP)	1997 images of bladder cancer 986 images of noncancer tissue	AUC of up to 0.99
Wu et al. [81]	Development of LNMDM	(AI-assisted workflow	998 patients	AUC from 0.978 to 0.998
Girard et al. [82]	Developing criteria to identify pelvic lymph node involvement in MIBC patients	ML-based combination of criteria	129 MIBC patients for training 44 patients for test	AUC of 0.59 in diagnostic performance compared to the experts (AUC = 0.64)
Gavriel et al. [84]	Development of an AI tool for predicting the 5-year prognosis of MIBC patients	ML-based algorithms	78 patients	71.4% accuracy in classification of patients who succumbed to MIBC
Nojima et al. [88]	Developing DLS as a diagnosis support tool for clinical cytology in urinary cytology	Deep Learning System (DLS)	Cytology images from Papanicolaou-stained urinary cytology glass slides obtained from 232 patients.	AUC of 0.9890 and an F1 score of 0.9002

## Data Availability

The data can be shared up on request.

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
