# Peer review of "Artificial Intelligence in Urooncology: What We Have and What We Expect"

_cancers, 2023, doi:10.3390/cancers15174282_

Round 1

Reviewer 1 Report

This paper presents a narrative review of artificial intelligence (AI) applications in urooncology, encompassing several urological cancers, including prostate cancer, kidney cancer, bladder cancer, upper tract urothelial carcinoma, and testicular tumors. The review plays a crucial role in summarizing the development of AI in urooncology, promoting synergistic advancements within the field.

The review extensively covers a significant number of research articles, thoroughly analyzing each paper with a clear and engaging writing style. The authors deserve praise for their diligent effort in collecting and comprehensively understanding all the papers.

Some comments regarding specific sections in the review:

- Sections 4.1 Imaging and 4.1 Diagnosis seem to share similar objectives, focusing on diagnosing prostate cancer using imaging modality and various predictors. Merging these subsections would enable a more critical review of research works related to prostate cancer diagnosis.

- Section 4.1 Follow-up cites articles that appear unrelated to AI, focusing solely on creating apps to enhance the follow-up rate of prostatectomy patients, without AI involvement.

- In section 4.2, "Renal cell carcinoma (RCC)" should be introduced when first mentioned in the article.

- It is common and commendable to summarize studies in a table in review papers. The author summarized the application of AI in prostate cancer only, but providing a summary table for all five types of cancers would enhance the reading experience.

- It is suggested to consider including or adopting figures from cited articles to aid the explanation. This is merely a suggestion, and the author may choose whether to incorporate them.

Minor comments:

1.       Line 22, NLP acronym is not defined as natural language processing. Suggest to add.

2.       "Materials and methods", the author did not go in detail about their inclusion/exclusion criteria. Additionally, in Line 54, the term "with the highest level of evidence and relevance" is vague. Suggest the author to explain more and add an inclusion/exclusion flowchart on how 225 articles became 79.

3.       Line 84, change conventional to convolutional.

4.       Line 88-89, neurons are not all necessarily fully connected in an ANN. Some neurons can be turned "off" to reduce overfitting.

5.       Line 128, change conventional to convolutional.

6.       Line 224, PSA acronym is not defined as prostate specific antigen. Suggest to add.

7.       Table 1 summarizes AI in prostate cancer but doesn’t include biomarkers and treatment, which are still related to prostate cancer. Additionally, author can add more details in the table such inputs/outputs to the model and what the model is verified against. This will add to the comprehensiveness of the review.  

8.       Line 244, change monitoring to monitored.

9.       The sentence is Lines 256-257 is redundant. Suggest to review or remove.

10.   "Follow up" section. This section is not really related to AI. IOMIC framework is simply an application development tool and not an algorithm.

11.   Line 388, what does the author mean by "acceptable"? This is subjective. Suggest to give a quantitative result.

12.   Line 498, DEG acronym is not defined as differentially expressesd gene. Suggest to add.

13.   Suggest the author to add two tables to summarize the AI findings in kidney cancer and bladder cancer, just like the one for prostate cancer. This will improve the flow of the review.

Ok, minor editing is required

Reviewer 2 Report

  • General: In the introduction, elaborate on how this literature review differs from previous reviews on this exact subject matter within the field. Additionally

  • General: Authors must proofread their works for grammatical and syntax errors throughout the paper.

  • Methods Section: Although “urooncology” is a search term used here, it would also make sense to have used “urologic oncology.”

  • Line 52: “kidney, cancer” is written, should this be written as “kidney cancer” without the comma instead?

  • In section 3. Definition and Types of AI: CNN is defined as “convolutional neural networks” However, in Figure 1, line 84, line 128, line 404 and line 413, CNN is mislabeled as a “conventional neural network.” 

  • Line 335: is a period or comma needed at the end of the sentence?

  • Line 652: The validity of this statement may need some verification and appears to be an overstatement without proper citation and evidence. The dawn of computing was in the 1950s, while AI in healthcare only entered in the 1970s. Potentially expanding the introduction of this review could strengthen the context and background in which you further expand to make this claim.

  • General: This study could benefit from standardizing subsections for each urologic cancer, ie Bladder, UTUC, and testicular cancer sections are not stratified by subsection (imaging, diagnosis, treatment, genomics, etc.) like other urologic cancers in the review. The purpose of the review should be to cover similar aspects that AI can be applied to in each of these subsections especially given other studies have done so. Additionally, this leads to a lackluster amount of disorganized ideas/information.

  • General: Subsections to each cancer could use introductory statements to privy the reader into each category of cancer and then subsequent subheaders.

  • General: The authors should find direct comparisons between AI modalities' prognostic/predictive capabilities vs. those of already existing validated nomograms. The authors should provide explicit rationale and examples of how AI improves upon these extant models. 

  • Authors should consider updating the search into 2023 to capture further studies to differentiate this study from previous reviews that have covered this topic extensively
  • References contain very few recent (2022) studies

Round 2

Reviewer 2 Report

Thank you to the authors for their diligent work on conducting significant revisions to improve this manuscript, including but not limited to expanding the literature review up through 2023, providing further context into the history and current role of AI, and clarifying the purpose of the review & how it may differ from prior studies.